# Identification and Bioinformatic Analysis of the *GmDOG1-Like* Family in Soybean and Investigation of Their Expression in Response to Gibberellic Acid and Abscisic Acid

**DOI:** 10.3390/plants9080937

**Published:** 2020-07-24

**Authors:** Yingzeng Yang, Chuan Zheng, Umashankar Chandrasekaran, Liang Yu, Chunyan Liu, Tian Pu, Xiaochun Wang, Junbo Du, Jiang Liu, Feng Yang, Taiwen Yong, Wenyu Yang, Weiguo Liu, Kai Shu

**Affiliations:** 1Institute of Ecological Agriculture, Sichuan Agricultural University, Chengdu 611130, China; yyz@nwpu.edu.cn (Y.Y.); chuanzheng618@nwpu.edu.cn (C.Z.); 71309@sicau.edu.cn (L.Y.); 71139@sicau.edu.cn (C.L.); TianPu@sicau.edu.cn (T.P.); xchwang@sicau.edu.cn (X.W.); junbodu@sicau.edu.cn (J.D.); jiangliu@sicau.edu.cn (J.L.); f.yang@sicau.edu.cn (F.Y.); yongtaiwen@sicau.edu.cn (T.Y.); mssiyangwy@sicau.edu.cn (W.Y.); 2School of Ecology and Environment, Northwestern Polytechnical University, Xi’an 710012, China; shankarc@nwpu.edu.cn

**Keywords:** soybean, seed dormancy, seed germination, phytohormone, *GmDOG1L* family

## Abstract

Seed germination is one of the most important stages during plant life cycle, and *DOG1* (*Delay of germination1*) plays a pivotal regulatory role in seed dormancy and germination. In this study, we have identified the *DOG1-Like* (*DOG1L*) family in soybean (*Glycine max*), a staple oil crop worldwide, and investigated their chromosomal distribution, structure and expression patterns. The results showed that the *GmDOG1L* family is composed of 40 members, which can be divided into six subgroups, according to their evolutionary relationship with other known *DOG1-Like* genes. These *GmDOG1Ls* are distributed on 18 of 20 chromosomes in the soybean genome and the number of exons for all the 40 *GmDOG1Ls* varied greatly. Members of the different subgroups possess a similar motif structure composition. qRT-PCR assay showed that the expression patterns of different *GmDOG1Ls* were significantly altered in various tissues, and some *GmDOG1Ls* expressed primarily in soybean seeds. Gibberellic acid (GA) remarkably inhibited the expression of most of *GmDOG1Ls*, whereas Abscisic acid (ABA) inhibited some of the *GmDOG1Ls* expression while promoting others. It is speculated that some *GmDOG1Ls* regulate seed dormancy and germination by directly or indirectly relating to ABA and GA pathways, with complex interaction networks. This study provides an important theoretical basis for further investigation about the regulatory roles of *GmDOG1L* family on soybean seed germination.

## 1. Introduction

Soybean is an important legume plant, and is now widely grown in many countries as a staple oil crop [1]. To meet the growing demand for soybean as food, oil and other resources, it is imperative to increase soybean production, and maize-soybean relay-intercropping is a model to increase soybean yield [2]. Seed germination is an important stage during the plant life cycle, which contributes to the spread and distribution of wild species and enhance the quality and yield of cultivated crops [3]. Most angiosperms begin a new stage of growth and development after seed dormancy, which is a very useful survival mechanism protecting plants from adverse environments [4,5]. Successful germination of seeds in the field is essential for the stable high yield [6]. Therefore, the intensive and extensive investigation of the regulatory mechanisms of seed dormancy and germination has an important scientific significance.

*DOG1* is a key gene in the control of seed dormancy and germination, which was first discovered through quantitative trait locus (QTL) analysis in *Arabidopsis* [7]. Following studies revealed that the amount of DOG1 protein regulates the seed dormancy level and germination capacity [8,9]. DOG1 starts to accumulate during seed maturation, and its protein structure is altered during the post-ripening stage. It has been reported that DOG1 express stably after maturation, and DOG1 protein abundance in freshly harvested seeds is important for seed dormancy release [8]. Interestingly, several studies have demonstrated that *DOG1* is highly conserved in diverse plant species, including *Lactuca sativa*, *Brassica rapa*, *Hordeum vulgare*, *Triticum aestivum* and *Oryza sativa* [10,11,12,13].

Phytohormone abscisic acid (ABA) and DOG1 protein are essential regulators of seed dormancy and germination. It is believed that *DOG1* is associated with ABA signaling pathway via clade A of type 2C protein phosphatases [14]. ABA is essential for DOG1 to function, and *DOG1* indirectly promotes the transcription of ABA biosynthesis gene *NCED* [8]. In addition, *AHG1* (*ABA-HYPERSENSITIVE GERMINATION 1*) and *AHG3*, other important components in ABA signaling pathway also act downstream of *DOG1* and are critical to *DOG1* function [14]. This indicated that *DOG1* controls seed germination by inhibiting the action of specific PP2Cs, which are the negative regulators of ABA signaling pathways.

The successful germination of soybean seeds is one of the preconditions required for a high yield. According to the previous studies, soybean is rich in oil and protein, which leads to the decrease of soybean germination rate, affecting the final yield [3]. Meanwhile, soybean seed germination is deficient, if exposed to diverse environmental stresses, such as salinity, drought or flooding stress, which further results in a serious decrease in final yield [15,16]. The studies of seed germination and dormancy are mainly focused on ABA and GA, and the molecular mechanisms of *DOG1* are extensively documented in *Arabidopsis* [7]. However, the studies of *DOG1* mediating the seed germination and dormancy are largely unknown especially in soybean cultivar, because of the shortage of the genetic information of *DOG1L* gene family in this species.

In this study, the characteristics and functions of *GmDOG1L* family are analyzed by several bioinformatics methods. The chromosome distribution, gene structure, gene expression patterns in different tissues and the responsiveness of phytohormone treatment of *GmDOG1L* family have been thoroughly investigated. We aimed for this study to help us in understanding the characteristics of soybean *DOG1* gene and its relationship with phytohormones.

## 2. Results

### 2.1. Identification of GmDOG1Ls Genes

The soybean-specific Hidden Markov Model for the DOG1 domain was used to identify *GmDOG1-Likes* (*GmDOG1Ls*), upon which a total of 40 non-redundant *GmDOG1-Like* candidates were identified in the soybean genome. Various data about these 40 *GmDOG1Ls*, including gene name, chromosome location, the number of exons and length of the coding sequences (CDS) were collected. As such, we named them *GmDOG1-Like-1* (*GmDOG1-L1*) to *GmDOG1-Like-40* (*GmDOG1-L40*) (Table 1). By using the NCBI CDD website, we found that the 40 *GmDOG1L* genes contained the conservative structural domain of *DOG1*(*PF14144*), which suggest that our analysis is reliable. Interestingly, among the 40 *GmDOG1Ls* identified, several members also contained the b-ZIP domain, presumably because of the three domains (PD870616, PD004114 and PD388003) conferring the characterization of *DOG1*, and relatively, PD004114 is also presented in the b-ZIP domain containing transcription factors described elsewhere [7].

The length of the coding sequences of the 40 *GmDOG1Ls* genes varied significantly, the longest one being *GmDOG1-L16* with 1554 bp, while the shortest one is *GmDOG1-L7* with only 285 bp. Accordingly, *GmDOG1-L16* had 517 amino acid residues, which was found to be the largest protein in the *GmDOG1L* family, while the smallest one was *GmDOG1-L7* with 94 amino acid residues. Furthermore, their predicted isoelectric points (pI) varied from 4.97 (*GmDOG1-L9*) to 9.97 (*GmDOG1-L12*) (Table 1).

### 2.2. GmDOG1L Genes Structure and Phylogenetic Analysis

In this study, we used the HMM method to search the *DOG1L* family in soybean, and as a result, a total of 40 *GmDOG1-like* genes were found. In addition, by using the same method, we found 20 *DOG1-like* genes in *Arabidopsis thaliana*, 20 *DOG1-like* genes in *Oryza sativa* and 53 *DOG1-like* genes in *Triticum aestivum*; the detailed gene names and gene IDs are shown in Appendix A. A total of 135 DOG1Ls from these five species were clustered into six subgroups (I–VI), including 40 *GmDOG1-Like*, 20 *AtDOG1-Like,* 53 *TaDOG1-Like (Triticum aestivum),* 20 *OsDOG1-Like (Oryza sativa)* and two *HvDOG1-Like (Hordeum vulgare)* (Figure 1). Among them, *AtDOG1* (*AT5G45830*), *AtDOG1-Like-1* (*AT4G18660*), *AtDOG1-Like-2* (*AT4G18680*), *AtDOG1-Like-3* (*AT4G18690*) and *AtDOG1-Like-4* (*AT4G18650*) were the first five discovered *DOG1* genes [7], followed by *HvDOG1-L1*, *HvDOG1-L2*, *TaDOG1-L1*, *TaDOG1-L4* and *OsDOG1-L3* for which a seed dormancy function has already been confirmed [13]. We found that *GmDOG1L* was included in all six subgroups. The six *DOG1-like* genes that have been proven to regulate seed dormancy were concentrated in the subgroups I and II. In addition, we found that *HvDOG1-2*, *GmDOG1-L2* and *GmDOG1-L26* were in the subgroup I alone. Meanwhile, *GmDOG1-L8*, *GmDOG1-L37* and *GmDOG1-L40* were found to be very close to *AtDOG1* in phylogenetic relationship, and *GmDOG1-L37* had the closest phylogenetic relationship with *AtDOG1*. From the phylogenetic tree analysis, we speculated that *GmDOG1-L8*, *GmDOG1-L37* and *GmDOG1-L40* might be the potential *DOG1* genes in soybean.

To narrow down the scope of the study, phylogenetic trees were constructed by selecting only *GmDOG1Ls*, five *AtDOG1Ls* genes and five other *DOG1-like* genes that have been proven to regulate seed dormancy [13] (Appendix A). Ten conserved DOG1Ls protein domains were characterized through MEME software analysis. As a result, we found that the motif composition of DOG1-Like in the same group had adequate consistency in phylogenetic trees (Figure 2). For example, in the subgroup IV, AtDOG1 consisted only of motif 1 and motif 8, with GmDOG1-L8, GmDOG1-L40 having the same composition. GmDOG1-L10, GmDOG1-L13 and GmDOG1-L11 all contain two motifs of AtDOG1. AtDOG1-L2, AtDOG1-L3 and AtDOG1-L4 contain motif 1 and motif 8. AtDOG1-L1 consisted of motif 5 and motif 8. These observations probably imply that motif 8 might be the most important component of DOG1. In addition, we found that TaDOG1-L1, TaDOG1-L4, HvDOG1-L1 and OsDOG1-L3 also contained motif 8. On the other hand, 37 of 40 contained both motif 1, whereas GmDOG1-L5, GmDOG1-L27 and GmDOG1-L37 were devoid of motif 1. While GmDOG1-L7 consisted of motif 1 alone. Interestingly, we found no any consistent motif between HvDOG1-L2 and other known DOG1-Like. It is to be noted that motif 8 was found in most of known DOG1 sequences, implying that motif 8 are the conserved structural domains shared by DOG1-like proteins and might play an important role. In summary, from the motif analysis, we speculate *GmDOG1-L8*, *GmDOG1-L10*, *GmDOG1-L13*, *GmDOG1-L11* and G*mDOG1-L40* as potential *DOG1* genes in soybeans. All of them had motifs contained in *AtDOG1*, and their motifs were similar to the motifs of three other *DOG1-Like* genes that had been identified to regulate seed dormancy.

The introns and exons of different *DOG1Ls* genes were varied as shown in the Appendix A. The smallest number of exons was only one, and the largest number was as many as 12. We found that *AtDOG1* contained three exons. *AtDOG1-L1*, *AtDOG1-L2*, *AtDOG1-L3*, *TaDOG1-L1*, *HvDOG1-L1* and *OsDOG1-L3* had only one exon, but *AtDOG1-L4* had two exons. In addition, the exons of *HvDOG1-L2* and *TaDOG1-L4* were temporarily unavailable. The distribution of exons and introns of *DOG1Ls* family was found to be complex, and even members of *DOG1Ls* that were grouped together in the evolutionary analysis had inconsistent exon composition.

In addition, we performed bioinformatics analysis on the DNA sequences of 40 *GmDOG1-Like* gene promoters, and the analysis results are shown in Appendix A. We analyzed the main role in *cis* element containing ABRE (*cis* acting element involved in the abscisic acid responsiveness), the TATC-box (*cis* acting element involved in gibberellin responsiveness), AuxRR-core (*cis* acting regulatory element involved in auxin responsiveness) and TGA-box (part of an auxin-responsive element), respectively. We found that *GmDOG1-L5*, *GmDOG1-L10*, *GmDOG1-L11*, *GmDOG1-L37* and *GmDOG1-L40* contained ABRE elements. *GmDOG1-L13* and *GmDOG1-L27* contained TGA elements with *GmDOG1-L28* containing ABRE and AuxRR elements. Interestingly, *GmDOG1-L37* contains three ABRE elements. This result suggested that these *GmDOG1Ls* might be involved in the regulation of phytohormone response in soybean.

### 2.3. Chromosomal Location and Gene Duplication

Forty *GmDOG1L* genes were distributed on 18 chromosomes of the soybean genome, except chromosomes 9 and 16 (Appendix A). Chromosome 1, 7, 8, 14 and 17 contained only one *GmDOG1L* gene. Chromosome 2, 4, 6, 11, 12, 18 and 19 each contained two *GmDOG1L* genes, whereas chromosome 3, 5, 15 and 20 each contained three *GmDOG1L* genes. Chromosome 10 contained four *GmDOG1L* genes, and chromosome 13 contained five *GmDOG1L* genes.

According to a previous research, the soybean genome experiences one whole genome triplication (WGT) event and two whole genome duplication (WGD) events with legume WGD and Glycine WGD, as well as about 75% of the genes in soybean, have multiple paralogs [17]. Among the paralog genes, 50% displayed expression of a sub-functionalization that may cause phenotypic variation [18,19]. In addition, dispersed duplicates generally arise by the transposition of DNA or RNA, which might play an important role in creating new genes and changing gene function [20,21]. Finally, we found that 24 of the 40 *GmDOG1Ls* were distributed in the duplication regions by using the MCScanX program, suggesting that these genes were generated by large-scale duplication events (Appendix A and Appendix A). Furthermore, gene family expansion might be caused by a tandem duplication event, generating consecutive copies of genes in the genome [22,23]. But no tandem duplications were detected in the *GmDOG1L* gene family.

We then calculated the nonsynonymous substitution rate (Ka) and synonymous substitution rate (Ks) of these duplicated gene pairs (Appendix A). In this study, only the Ka/Ks ratio of GmDOG1-L2 and GmDOG1-L1, GmDOG1-L16 & GmDOG1-L3 and GmDOG1-L32 and GmDOG1-L25 was greater than 1, which is considered to be subject of positive selection, the remaining Ka/Ks ratios were found to be less than 1, which is considered as a purification selection [24]. In addition, the Ks value of GmDOG1-L2 and GmDOG1-L1, GmDOG1-L10 and GmDOG1-L34, GmDOG1-L10 and GmDOG1-L21, GmDOG1-L15 and GmDOG1-L34, GmDOG1-L16 and GmDOG1-L3 and GmDOG1-L32 and GmDOG1-L25 were found to be lesser than 1.3 and greater than 0.3, suggesting that their divergence time was after legume WGD event and before the Glycine WGD. The Ks value of other duplicated gene pairs were less than 0.3, which indicates that their divergence time was after the Glycine whole genome duplication (WGD) event [17,25].

### 2.4. Expression Profiles of GmDOG1L Genes in Various Tissues of Soybean

In order to understand the roles of *GmDOG1Ls* in the growth and development of soybean, we selected 10 *GmDOG1L* genes from different subgroups for tissue specific expression analysis. First, we selected the *GmDOG1-L37* gene from the subgroup II, which was found closest to *AtDOG1* in phylogenetic analysis among the 40 members of *GmDOG1Ls*, and *GmDOG1-L37* contained three ABRE elements. Secondly, we selected the *GmDOG1-L11* gene from the subgroup II, as *GmDOG1-L11* contained *AtDOG1*’s motif composition along with containing ABRE elements. Third, we chose *GmDOG1-L27* from the subgroup II, as it was close to *HvDOG1-L1* and *TaDOG1-L1*, and *GmDOG1-L27* also contained TGA elements. Finally, *GmDOG1-L10* was also employed for its closest evolutionary relationship to *AtDOG1-L3* from the subgroup II, which had the similar motifs composition of *AtDOG1*, and *GmDOG1-L10* contained ABRE elements. In addition, considering that soybean genome experiences one whole genome triplication (WGT) event and two whole genome duplication (WGD) events, functional differentiation might have occurred in some *GmDOG1Ls*. In order to explore whether *GmDOG1Ls* had any functional differentiation, we chose *GmDOG1-L2* and *GmDOG1-L26* from subgroup I, *GmDOG1-L39* from subgroup III, *GmDOG1-L30* from subgroup IV, *GmDOG1-L3* from subgroup V and *GmDOG1-L1* from subgroup VI. These 10 *GmDOG1L* genes were selected from different subgroups as representatives to study whether there were any differences in expression patterns of *GmDOG1Ls* in different various tissues.

The qRT-PCR assay was employed to investigate the expression patterns of these *GmDOG1Ls* in several soybean tissues, including root, stem, leaf, flower, apical meristems, pod, developing seed and dry seed. The results showed that all the genes were expressed in eight tissues with different levels (Figure 3). Interestingly, the expression of *GmDOG1-L1*, *GmDOG1-L2*, *GmDOG1-L3* and *GmDOG1-L39* was found to be highest in the pods, followed by dry seeds. *GmDOG1-L11, GmDOG1-L27, GmDOG1-L30* and *GmDOG1-L37* were most highly expressed in dry seeds. Furthermore, expression of *GmDOG1-L26* was unexpected with high expression level in leaves, followed by pods and dry seeds. In addition, in the phylogenetic analysis, *GmDOG1-L26* and *GmDOG1-L2* were in subgroup I, but there were significant differences in gene expression in different soybean tissues. Finally, *GmDOG1-L26*, *GmDOG1-L27* and *GmDOG1-L30* genes were found to be highly expressed in developing seeds.

These 10 *GmDOG1L* genes were all expressed relatively high in seeds, which is consistent with the previous studies reporting that *DOG1* is a key gene regulating seed dormancy and germination. In general, *GmDOG1-L11, GmDOG1-L27, GmDOG1-L30* and *GmDOG1-L37* were indeed primarily expressed in dry seeds indicating that these genes might mainly function in seed biology. In summary, from tissue specific expression analysis, we believed that *GmDOG1-L11, GmDOG1-L27, GmDOG1-L30* and *GmDOG1-L37* might be the potential *GmDOG1* genes.

Since some specific primers of *GmDOG1Ls* members cannot be designed, we obtained transcriptome data of soybean from NCBI GEO DataSets, which contained various organs and different tissues (GEO accession: GSE123655). We extracted 40 *GmDOG1Ls* expression data from the transcriptome data and used TBtools software to draw the clustering heatmap (Figure 4). We found that *GmDOG1-L4*, *GmDOG1-L19*, *GmDOG1-L25*, *GmDOG1-L27*, *GmDOG1-L32* and *GmDOG1-L40* were highly expressed in all tissues, implying that these genes might be important throughout the soybean life cycle. In addition, the transcription expression of *GmDOG1-L4, GmDOG1-L5, GmDOG1-L10*, *GmDOG1-L19, GmDOG1-L25, GmDOG1-L32* and *GmDOG1-L40* increased from ISC (immature seed coat-Stage: 5–6 mg), ISC2 (immature seed coat-Stage: 25–50 mg) to ISC3 (immature seed coat-Stage: 200–300 mg), but decreased at the MSC (mature seed coat) stage. Meanwhile, transcription expression of *GmDOG1-L3*, *GmDOG1-L6, GmDOG1-L12*, *GmDOG1-L16, GmDOG1-L27, GmDOG1-L36* and *GmDOG1-L37* gradually increased from ISC, ISC2 and ISC3 to MSC. They gradually increased their expression during seed development period, thus, we speculated that these genes might be the potential *GmDOG1* genes.

### 2.5. Expression Analysis of GmDOG1L under Phytohormones Treatment

To further investigate the relationship between *GmDOG1Ls* and phytohormones, seven *GmDOG1L* genes were selected from different groups for qRT-PCR assay. Compared with control (CK), the expression of *GmDOG1-L10, GmDOG1-L27* and *GmDOG1-L30* were up-regulated, while the transcription level of *GmDOG1-L11, GmDOG1-L26, GmDOG1-L37* and *GmDOG1-L39* were down-regulated with distinct levels, after exogenous ABA treatment (Figure 5). Interestingly, the expression of all the *GmDOG1L* genes was down-regulated under GA treatment, suggesting that the expression of these *GmDOG1Ls* was inhibited by GA. In addition, the expression of *GmDOG1-L10*, *GmDOG1-L11* and *GmDOG1-L30* were up-regulated under FL (fluridone) treatment, while the expression of *GmDOG1-L26*, *GmDOG1-L27*, *GmDOG1-L37* and *GmDOG1-L39* were down-regulated. Intriguingly, the expression of all *GmDOG1L* genes were down-regulated under PAC (paclobutrazol) treatment.

## 3. Discussions

*DOG1s* is a key gene regulating seed dormancy and germination [7]. Compared with wild type, *dog1* mutant seeds germinated much faster, while the germination was delayed in overexpressed *AtDOG1* seeds [7]. In this study, we used the HMM approach to look out for *DOG1-like* genes in soybeans, and finally found 40 *GmDOG1Ls*. We performed phylogenetic analysis with 40 *GmDOG1-like* genes, 20 *AtDOG1-Like* genes, 20 *OsDOG1-like* genes, 53 *TaDOG1-Like* genes and two *HvDOG1-like* genes. From the phylogenetic analysis, *GmDOG1-L37* with *AtDOG1* were the closest, followed by *GmDOG1-L8* and *GmDOG1-L40.* Moreover, *GmDOG1-L37* was found in between *AtDOG1* and *OsDOG1-L3*, which is one of the genes proven to regulate seed germination. The 135 *DOG1-like* genes were divided into six subgroups, and we found that *AtDOG1* and 5 *DOG1-like* genes that have been proven to regulate seed dormancy were clustered in the subgroup I and II, which might indicate that these two subgroups played a special role in the phylogenetic analysis. In addition, among the remaining four subgroups, no *DOG1-like* gene that has been confirmed to regulate seed dormancy was found, which might imply that the *DOG1-like* genes of these four subgroups are not potential *DOG1s* and might be those that have lost the ability to regulate seed dormancy.

In Figure 2, the motifs of *DOG1Ls* had obvious characters in the same subgroup, whereas in Appendix A, the exons of *DOG1Ls* had no obvious characters. Previous investigations have shown that gene duplication is an important approach to generate new genes, which is actually a protection mechanism for plants to adapt to changing environment [26,27]. In this study, we found that most of the *GmDOG1Ls* were distributed in duplication blocks, indicating that WGD or segmental duplications plays an important role in the extension of *GmDOG1L* family [28]. We speculate that the increase in the number of *GmDOG1L* gene family might have been primarily caused by gene duplication, especially the WGD.

We chose 10 *GmDOG1Ls* that were relatively highly expressed in dry seeds (Figure 3). Among the 10 genes, *GmDOG1-L11*, *GmDOG1-L27, GmDOG1-L30* and *GmDOG1-L37* showed the highest expression in dry seeds, thus, we speculated them as the potential *DOG1* genes in soybean. After exploring the genes selected further, we found that the expression patterns of the genes in the same subgroup showed similar patterns. For example, *GmDOG1-L11* and *GmDOG1-L27* genes from the group II were highly expressed in dry seeds. *GmDOG1-L26* and *GmDOG1-L30* had similar motif compositions, and the expression patterns were also similar, in parallel to their high expression in leaves. These tissue-specific expression results suggest that some of the 40 *GmDOG1L* family members were not potential *GmDOG1* genes, thus showing different biological functions. In addition, by analyzing the transcriptome data, we found that the expression of *GmDOG1-L3*, *GmDOG1-L6, GmDOG1-L12*, *GmDOG1-L16, GmDOG1-L27, GmDOG1-L32, GmDOG1-L36* and *GmDOG1-L37* genes were sustained during seed development. Therefore, we speculate that they might be the potential *DOG1* genes in soybean.

Phytohormones ABA and GA play a leading role in regulating seed dormancy and germination, and they antagonistically mediate diverse plant developmental processes including seed dormancy and germination [29,30]. For example, when seeds are treated with ABA, dormancy is enhanced and germination is inhibited [6]. Contrarily, when seeds are treated with GA, seed dormancy is inhibited and seed germination is promoted [6]. Analysis of qRT-PCR of *GmDOG1Ls* under phytohormone treatment showed that the precise correlation between *GmDOG1Ls* expression and ABA was not detected as ABA inhibits some of the *GmDOG1Ls* expression while promoting others. Similarly, the correlation between FL and *GmDOG1Ls* was observed to be the same. In addition, experimental results showed that GA and PAC down-regulated the expression of *GmDOG1L* genes. The expression of these seven genes had no obvious characters under phytohormone treatment. From this, we speculated that some members of the 40 *GmDOG1L* may not belong to the *GmDOG1* family, and they might not have the function of regulating seed dormancy and germination, thus, their response to phytohormones is inconsistent with the results obtained.

## 4. Conclusions

In this study, we performed a comprehensive analysis of *GmDOG1L* gene family, providing a perspective for the evolution of this family. Using *DOG1* HMM, we identified 40 *GmDOG1Ls* from soybean genome. These *GmDOG1Ls* were distributed on 18 chromosomes of soybean and were divided into five subgroups according to their evolutionary relationship. We found that *GmDOG1Ls*’ motifs are similar. Gene duplication analysis indicated that the WGD or segmental duplications might lead to the expansion of *GmDOG1L* family. Expression profile analysis showed that *GmDOG1-L11*, *GmDOG1-L27, GmDOG1-L30* and *GmDOG1-L37* had the highest expression in seeds compared to other *GmDOG1Ls*. The precise correlation between *GmDOG1L* genes expression and ABA or FL was not detected, but the expression of *GmDOG1Ls* was inhibited under GA and PAC treatments. More importantly, we identified some potentially useful *GmDOG1L* genes through this study, such as *GmDOG1-L11, GmDOG1-L37* and *GmDOG1-L40*. Not only were they found closer to identified *DOG1-like* in phylogenetic tree analysis, but their motif compositions were also similar to identified *DOG1-like* compositions, and their expression in seeds was the highest in qRT-PCR assay. Among them, we believe that *GmDOG1-L37* is the most likely gene to become potential *GmDOG1*. It was not only found closest to *AtDOG1* in phylogenetic analysis, but also had the highest expression in seeds in qRT-PCR analysis. Transcriptome data also confirmed that its expression continues to increase during seed development and their promoter also contained three ABRE elements. Therefore, we believe that *GmDOG1-L37* might have the greatest potential to become *GmDOG1* gene in soybean. These results provide a basis for further understanding the molecular functions of *GmDOG1L* family and the specific mechanisms of *GmDOG1L* family in regulating seed dormancy and germination.

## 5. Materials and Methods

### 5.1. Identification of GmDOG1Ls

The soybean genome and protein sequences were downloaded from Phytozome12 website (http://www.phytozome.net/). The Hidden Markov Model (HMM) of conservative structure domain of DOG1 (PF14144) was downloaded from the PFAM database (http://pfam.xfam.org/) [13,31,32]. Predicted GmDOG1Ls were scanned with HMMER 3.1 software using the HMM of conserved domain of DOG1 [33,34]. We then used those protein sequences (E-value < 0.001) to construct a new HMM model of soybean using HMMER 3.1 software. This new soybean-specific HMM was used to identify *GmDOG1Ls* (E-value < 0.001). In order to ensure the accuracy of the results, 40 GmDOG1Ls proteins sequence were submitted on the NCBI website (https://www.ncbi.nlm.nih.gov/Structure/cdd/wrpsb.cgi) for further verification [35].

### 5.2. Chromosomal Distribution, Gene Structure and Phylogenetic Analysis

The MapGene2Chrom (http://mg2c.iask.in/mg2c_v2.0/) website was used to draw a visual distribution map of *GmDOG1Ls* on the chromosomes [36]. The exon and intron distribution patterns of *GmDOG1Ls* were analyzed by Gene Structure Display Server (GSDS2.0: http://gsds.cbi.pku.edu.cn/) [37]. Conserved motifs of GmDOG1Ls were predicted by MEME 5.0.5 software (search parameters: Maximum Number of Motifs:10; Motif E-value Threshold: no limit; Minimum Motif Width: 6; Maximum Motif Width: 50; Minimum Sites per Motif: 2; Maximum Sites per Motif: 50) [38]. Multiple sequence alignments of GmDOG1Ls were analyzed by ClustalW [39]. Phylogenetic analysis was conducted with MEGA 7 software [40]. We used maximum likelihood method to infer the evolutionary relationship of GmDOG1Ls [41]. The bootstrap consensus tree was repeated 1000 times [42]. Figtree 3.1 software was used to analyze and garnish the phylogenetic tree. The theoretical isoelectric point (pI) and molecular weight (MW) of GmDOG1Ls were predicted through ExPASy proteomics server (https://web.expasy.org/protparam/) [43].

### 5.3. Synteny Analysis of GmDOG1Ls

Duplication events for *GmDOG1Ls* was performed using the Multiple Collinearity Scan toolkit X (MCScanX) program with the default parameters [44]. According to the results, KaKs-Calculator software was used to calculate the non-synonymous replacement rate (Ka) and synonymous replacement rate (Ks) of duplication genes [45]. We used the Circos tool to draw the positions of genes and segmental duplicated regions on the soybean chromosomes [46,47].

### 5.4. Gene Expression Analysis

As previously described, we performed the total RNA preparation and first-strand cDNA synthesis as well as a qRT-PCR assay [48]. Total RNA was treated with DNase I, and then 2 μg total RNA was reverse-transcribed using Moloney murine leukemia virus reverse transcriptase (200 units per reaction; Promega Corporation), according to the manufacturer’s protocol. The soybean housekeeping gene *GmTubulin* was used as endogenous reference gene, and each reactions were repeated three times [3].

The qRT-PCR reaction system was 10 μL, which included: 0.4 μL forward primer and reverse primer, 3.6 μL DNase-free ddH_2_O, 1 μL cDNA and 5 μL Vazyme™ AceQ qPCR SYBR Green Mastermix. The qRT-PCR reaction procedure was set as follows: 94 °C for 2 min 30 s, and then 40 cycles of 94 °C for 10 s and 60 °C for 32 s. Each experiment values represent three biological replicates. The qRT-PCR performed using Vazyme™ AceQ qPCR SYBR Green Master mix on a QuantStudio 6 Flex Real-Time PCR System (Thermo Fisher Scientific, USA) [3]. The expression level of *GmDOG1Ls* were calculated by the comparative C_T_ method [49]. Online primers were designed using NCBI primer design tool site (https://www.ncbi.nlm.nih.gov/tools/primer-blast/index.cgi?LINK_LOC=BlastHome). The detailed information of the primers is shown in Appendix A.

## Figures and Tables

**Figure 1 plants-09-00937-f001:**
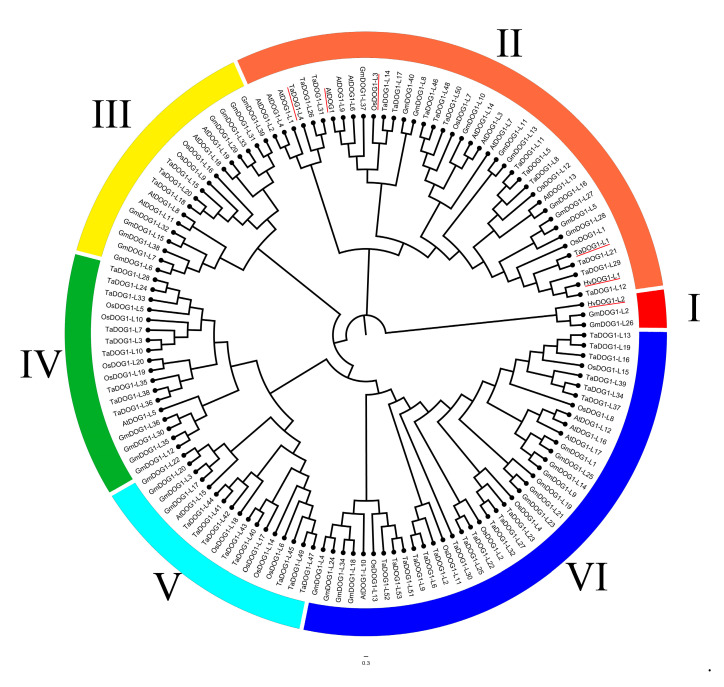
Phylogenetic analysis of *DOG1Ls* from *Glycine max*, *Arabidopsis, Hordeum vulgare*, *Triticum aestivum* and *Oryza sativa*. *DOG1-like* genes for which a seed dormancy function has been confirmed are red underlined. Clustal W software was used to align 135 *DOG1Ls*. A maximum likelihood phylogenetic tree was constructed by MEGA7.0 software with the Poisson model and 1000 bootstrap replications. The group I–VI use different color difference between them.

**Figure 2 plants-09-00937-f002:**
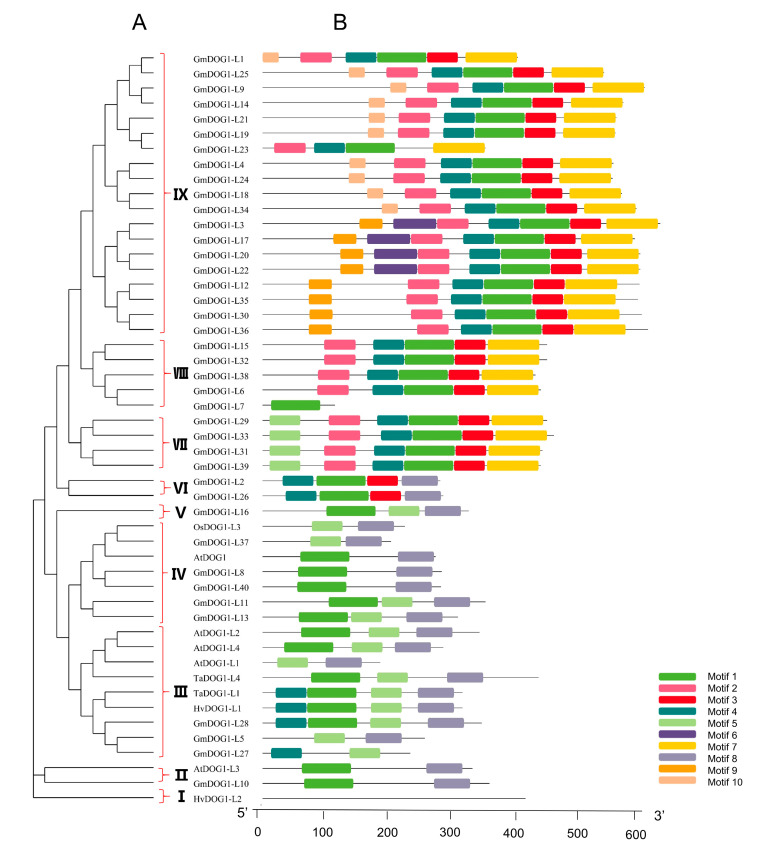
Phylogenetic analysis and conserved motifs of *DOG1Ls.* (**A**) The phylogenetic relationship of *DOG1Ls.* A maximum likelihood phylogenetic tree was constructed by MEGA7.0 software with the Poisson model and 1000 bootstrap replications. (**B**) Conserved motif arrangements of DOG1Ls. Ten conserved motifs labeled with different colors were found in the DOG1Ls sequences using the MEME program.

**Figure 3 plants-09-00937-f003:**
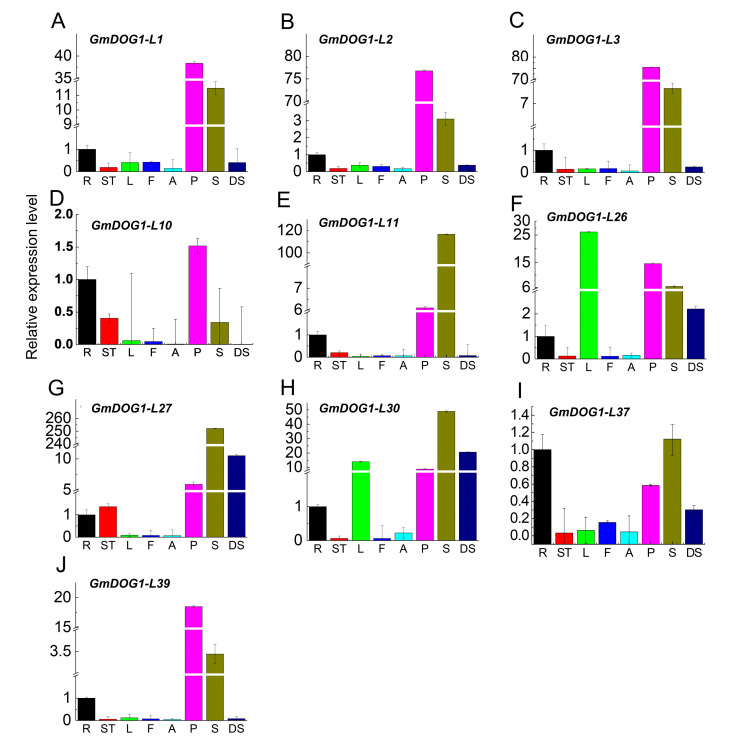
Expression profiles of *GmDOG1Ls* in different tissues (**A**–**J**). Expression levels of 10 selected *GmDOG1Ls* were examined by qRT-PCR analysis. The housekeeping gene *GmTubulin* was used as an endogenous reference gene. Roots (R), stems (ST), leaves (L), flowers (F), apical meristems (A), pods (P), dry seeds (S) and developing seeds (DS) were sampled for qRT-PCR analysis. Error bars represent standard errors.

**Figure 4 plants-09-00937-f004:**
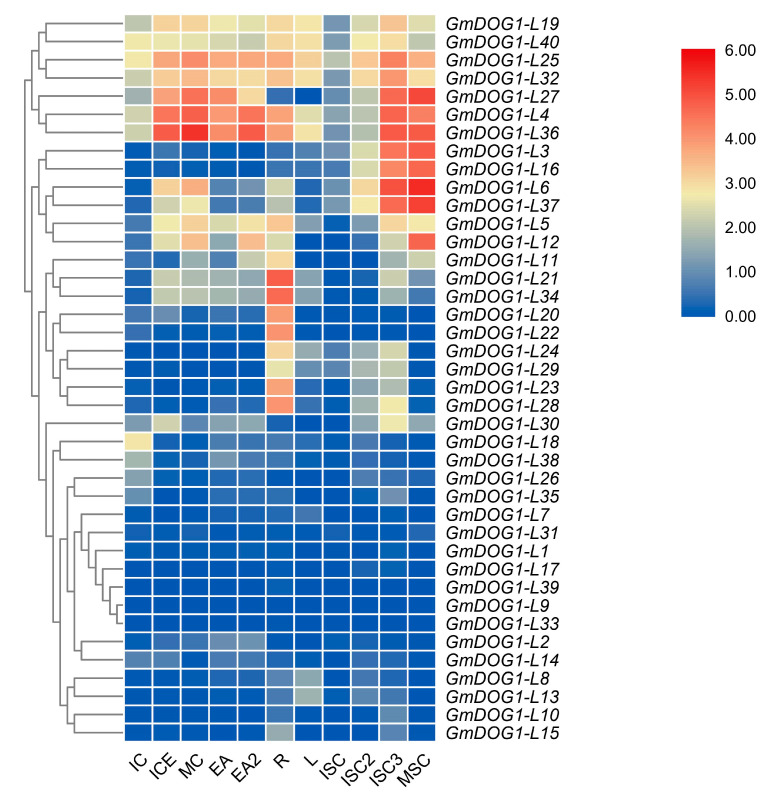
Hierarchical clustering of *GmDOG1Ls* transcriptome expression profiles including 11 samples at different tissue and developmental stages. Immature cotyledon (IC), immature cotyledon without embryo axis (ICE), mature cotyledon (MC), embryo axis-Stage: 300–400 mg (EA), embryo axis-Stage: 200–300 mg (EA2), roots (R), leaves (L), immature seed coat-Stage: 5–6 mg (ISC), immature seed coat-Stage: 25–50 mg (ISC2), immature seed coat-Stage: 200–300 mg (ISC3) and mature seed coat (MSC) were sampled for RNA-seq analysis.

**Figure 5 plants-09-00937-f005:**
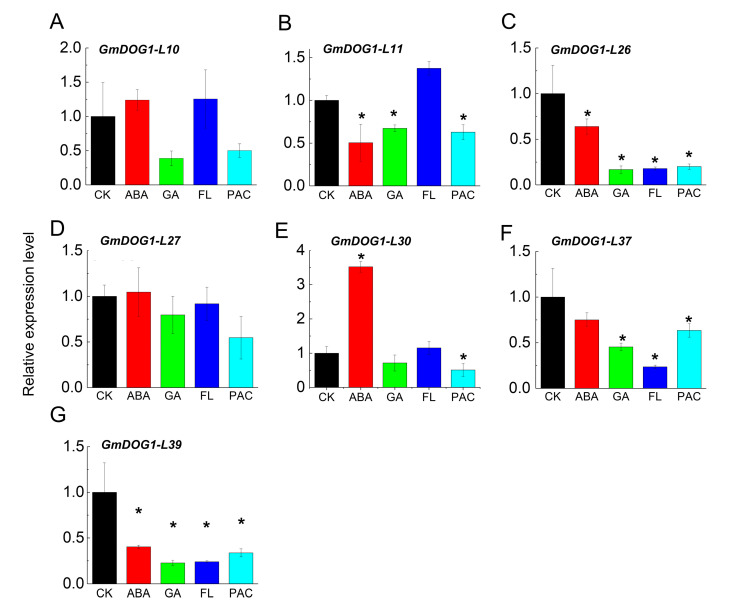
Expression profiles of *GmDOG1Ls* in soybean seeds after phytohormone treatment (**A**–**G**). Phytohormones include Abscisic Acid (ABA), Gibberellin (GA), FL and PAC, they all have a concentration of 5 μmol/L. Expression levels of seven selected *GmDOG1Ls* were examined by qRT-PCR analysis. The housekeeping *GmTubulin* was used as an endogenous reference gene. The asterisk (*) indicates a significant difference at P < 0.05 by Student’s t-test analysis.

**Table 1 plants-09-00937-t001:** Characterization of the *GmDOGL1s* family in soybean.

Gene Name	Locus ID	CDS (bp)	Chromosome Number	Number of Exons	Length (aa)	pI	MW (Da)
*GmDOG1-L1*	Glyma.01G084200	1461	1	12	486	7.32	54309.28
*GmDOG1-L2*	Glyma.02G097900	1404	2	11	467	7.32	51853.46
*GmDOG1-L3*	Glyma.02G176800	1455	2	11	484	7.77	53927.08
*GmDOG1-L4*	Glyma.03G127600	1383	3	11	460	7	50928.00
*GmDOG1-L5*	Glyma.03G128200	873	3	7	290	8.82	32365.88
*GmDOG1-L6*	Glyma.03G142400	1476	3	11	491	8.22	55200.29
*GmDOG1-L7*	Glyma.04G151800	285	4	2	94	5.04	10511.23
*GmDOG1-L8*	Glyma.04G254800	1089	4	8	362	8.69	40929.00
*GmDOG1-L9*	Glyma.05G113200	579	5	2	192	4.97	22591.38
*GmDOG1-L10*	Glyma.05G182500	1113	5	8	370	7.11	42096.79
*GmDOG1-L11*	Glyma.05G195200	858	5	1	285	5.28	32592.96
*GmDOG1-L12*	Glyma.06G090900	495	6	2	164	9.97	18889.84
*GmDOG1-L13*	Glyma.06G107300	1068	6	8	355	6.21	39984.68
*GmDOG1-L14*	Glyma.07G151600	702	7	1	233	5.32	26409.83
*GmDOG1-L15*	Glyma.08G140100	1140	8	8	379	7.78	43105.04
*GmDOG1-L16*	Glyma.10G092100	1554	10	10	517	6.61	57921.5
*GmDOG1-L17*	Glyma.10G194800	696	10	4	231	7	26599.48
*GmDOG1-L18*	Glyma.10G276100	1371	10	11	456	6.08	50993.85
*GmDOG1-L19*	Glyma.10G296200	999	10	8	332	8.94	37141.79
*GmDOG1-L20*	Glyma.11G183700	1482	11	12	493	5.87	54385.80
*GmDOG1-L21*	Glyma.11G236300	1095	11	8	364	6.28	41178.84
*GmDOG1-L22*	Glyma.12G088700	1506	12	12	501	6.61	55253.16
*GmDOG1-L23*	Glyma.12G184500	1467	12	12	488	6.75	54118.97
*GmDOG1-L24*	Glyma.13G085100	1113	13	8	370	7.13	41915.73
*GmDOG1-L25*	Glyma.13G193700	1410	13	11	469	5.98	51989.98
*GmDOG1-L26*	Glyma.13G228600	765	13	1	254	8.70	28971.62
*GmDOG1-L27*	Glyma.13G269500	807	13	2	268	5.86	30487.66
*GmDOG1-L28*	Glyma.13G316900	1473	13	12	490	6.96	54630.43
*GmDOG1-L29*	Glyma.14G167000	1113	14	8	370	8.26	41888.77
*GmDOG1-L30*	Glyma.15G083900	873	15	1	290	9.37	32967.42
*GmDOG1-L31*	Glyma.15G182700	888	15	2	295	6.71	33829.21
*GmDOG1-L32*	Glyma.15G232000	1494	15	11	497	6.1	55125.35
*GmDOG1-L33*	Glyma.17G154000	636	17	3	211	5.95	24440.01
*GmDOG1-L34*	Glyma.18G020900	1089	18	8	362	7.78	41052.71
*GmDOG1-L35*	Glyma.18G202900	699	18	1	232	5.72	26078.56
*GmDOG1-L36*	Glyma.19G130200	1380	19	11	459	8.53	50717.82
*GmDOG1-L37*	Glyma.19G145300	1476	19	11	491	8.71	55358.4
*GmDOG1-L38*	Glyma.20G113600	1368	20	11	455	5.98	50671.73
*GmDOG1-L39*	Glyma.20G195100	708	20	4	235	8.2	26896.70
*GmDOG1-L40*	Glyma.20G246400	1335	20	11	444	5.91	49505.41

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
