# Peer review of "Identification and Bioinformatic Analysis of the *GmDOG1-Like* Family in Soybean and Investigation of Their Expression in Response to Gibberellic Acid and Abscisic Acid"

_plants, 2020, doi:10.3390/plants9080937_

Round 1

Reviewer 1 Report

This research article identifies and describes the gene family for soybean DOG1-like genes and characterizes gene expression levels for select family members. DOG1 plays a role in regulating seed dormancy and germination in multiple plant species, so DOG1 genes in soybean might be targets for manipulation in crop improvement efforts. Most of the work reported here is straightforward and was performed and reported correctly.

My main concern with the paper lies in overstating the relevance and completeness of the work and heavy speculation in the discussion. The title suggests a characterization of the DOG1-like family, but that would typically indicate more than bioinformatic analysis. I might suggest a title like: “Identification and bioinformatic analysis of the GmDOG1-Like family in soybean and examination of their expression in response to gibberellic acid and abscisic acid”.  There were also a number of cases in the paper that exaggerated the usefulness of the study:

-Line 80: these results… “can regulate seed dormancy and germination by modifying GOD1Ls?” The sentence does not make sense as written and the claim is not supported.

For the bioinformatic analyses, was only the dominant transcript for each gene used and how was that determined? How many genes have evidence for multiple transcript variants?

All of the 5 non-Arabidopsis DOG1-like genes from other species have been confirmed to regulate seed dormancy using genetic tests (knockout mutants)? TaDOG1-L1, TaDOG1-L4, HvDOG1-L1, HvDOG1-L2, and OsDOG1-L3? Line 111. And also all 5 Arabidopsis genes? Line 279.

The phylogenetic analysis would be more useful if all of the DOG1-like genes from other species were included (Arabidopsis, Barley, Wheat, Rice, and maybe another dicot). Inclusion of only genes that have been genetically characterized limits how the data can be interpreted. If a broader phylogenetic analysis could be done, the entire paper would be strengthened. It would not be necessary to do the motif or promoter or exon analysis on all the genes from other species.

Line 137: When you say motif 8 might be the most important, what do you mean? Most conserved? Important for what kind of function? Has this motif been characterized in other species?

In selection of the 8 genes to test expression in various tissues, GmDOG1-L11 was chosen as closest in “evolutionary relation” to AtDOG1, but the closest in phylogeny (which show evolutionary relation) are L3 and L37. This should be rephrased as L11 was chosen as a likely candidate for DOG1 function based on motif and promoter analyses.

It is very unfortunate that qPCR data could not be gained for L8 and L40, since they are good candidates for DOG1 function. If these could be included, the completeness of the study would be much higher. What about L37?

What was the rationale for examining different genes in the various tissue analysis vs the GA/ABA treatment anlysis? L10 and L37 were only examined in hormone analysis while L1,L2,L3 were only examined in the various tissue. It would be much better to include all of the genes in all of the analyses. Was there missing data or did the data not look good? The current presentation makes it seem like only selective data was shown.

The discussion of green-stay G gene could be excluded. It would be highly unusual for it to happen to be one of the DOG1-related genes.

In the Introduction, Explain why seed germination is a particularly important area to study for soybeans.

Overly speculative: Line 311 (discovery… seems to provide a new green pathway…??); Line 320 (..family has undergone obvious functional differentiation.); Line 321 (some of the DOG1L genes have lost the function…); Line 330 (DOG1Ls might have undergone functional differentiation). Those statements are not supported and should be re-worded or removed. The data reported here does not support speculation about the function of the genes, functional differentiation, etc.). The various responses to ABA treatment do not necessarily mean that their biochemical or metabolic functions have diverged.

English editing is needed before publication.

Minor comments/suggestions:

Line 42: what is meant by “the rational”?

Line 66: Probably it is not worth noting that China is a large soybean producer. This is the only line in the paper about China and it seems out of place.

Line 106: reference to “these 5 species” occurs before the species are introduced.

Line 122: “closest” to what?

Line 159: What do you mean that the exon are temporarily unavailable? Is that an error message from a webtool?

I would suggest moving Figure 3 to supplemental. The phylogeny in Fig. 3 is redundant with Fig. 2. If you do no want to move Fig. 3 to supplemental, maybe think of a way to combine Fig 2 and Fig 3 into a single figure?

Line 173, 174: Clarify that those genes contain recognizable promoter elements. The failure to identify does not necessarily mean the are absent right?

Line 192: How common are tandem duplication in soybean? Is it unusual that you did no identify any?

Was primer specificity tested by sequencing PCR products and examining melt curves?

Line 244: Be specific about what “other analyses” you are referring to.

Line 264: “both” should be “all”?

1st paragraph Discussion could be left out.

Line 295: the eight genes were chosen because they were highly expressed? Was this based on existing transcriptomics data?

Line 327: “We found that the gene motifs… is consistent.” Does not form a complete sentence. What is meant by this?

Reviewer 2 Report

Some abbreviations should be explained e.g.

DOG (DELAY OF GERMINATION)

paragraph from lines 65 to 73 seems to be at the beginning of the introduction

the purpose of the work should be presented clearly in the paragraph from lines 74 to 81

what soybean genome was used for the analysis - was it the Williams82 genome

throughout the publication, gene names should be written the same way,

GmDOG1-L1

line 201 and 202 : should be Glycine;

Round 2

Reviewer 1 Report

Most of my concerns and suggestions have been addressed. There remains some awkward English usage that should be fixed prior to publication.

The expanded phylogenetic analysis does not provide useful information as presented currently. My suggestion was to include all DOG1-like genes from a few species, not a single DOG1-like homolog from many different species. To understand the gene family using phylogeny, it would be best to analyze the entire gene family from a few select species (maybe Arabidopsis, soybean, rice, wheat). That analysis would show whether the previously-characterized DOG1-like genes grouped together and how many subgroups were present in multiple species.
